# An ancient yet flexible *cis*-regulatory architecture allows localized Hedgehog tuning by *patched/Ptch1*

David S Lorberbaum[1,2], Andrea I Ramos[1,2], Kevin A Peterson[3,4], Brandon S Carpenter[1], David S Parker[1†], Sandip De[5], Lauren E Hillers[1‡], Victoria M Blake[1,5], Yuichi Nishi[6], Matthew R McFarlane[3§], Ason CY Chiang[1], Judith A Kassis[5], Benjamin L Allen[1], Andrew P McMahon[3,6], Scott Barolo[1*]

[1]Department of Cell and Developmental Biology, University of Michigan Medical School, Ann Arbor, United States; [2]Program in Cellular and Molecular Biology, University Of Michigan Medical School, Ann Arbor, United States; [3]Department of Stem Cell and Regenerative Biology, Harvard University, Cambridge, United States; [4]The Jackson Laboratory, Bar Harbor, United States; [5]Program in Genomics of Differentiation, *Eunice Kennedy Shriver* National Institute of Child Health and Human Development, National Institutes of Health, Bethesda, United States; [6]Department of Stem Cell Biology and Regenerative Medicine, Eli and Edythe Broad Center for Regenerative Medicine and Stem Cell Research, University of Southern California Keck School of Medicine, Los Angeles, United States

*For correspondence: sbarolo@umich.edu

Present address: †Department of Biology, Elon University, Elon, United States; ‡Graduate Program in Cellular and Molecular Biology, University of Wisconsin, Madison, United States; § Department of Molecular Genetics, University of Texas Southwestern Medical Center, Dallas, United States

**Competing interests:** The authors declare that no competing interests exist.

**Abstract** The Hedgehog signaling pathway is part of the ancient developmental-evolutionary animal toolkit. Frequently co-opted to pattern new structures, the pathway is conserved among eumetazoans yet flexible and pleiotropic in its effects. The Hedgehog receptor, Patched, is transcriptionally activated by Hedgehog, providing essential negative feedback in all tissues. Our locus-wide dissections of the *cis*-regulatory landscapes of fly *patched* and mouse *Ptch1* reveal abundant, diverse enhancers with stage- and tissue-specific expression patterns. The seemingly simple, constitutive Hedgehog response of *patched/Ptch1* is driven by a complex regulatory architecture, with batteries of context-specific enhancers engaged in promoter-specific interactions to tune signaling individually in each tissue, without disturbing patterning elsewhere. This structure—one of the oldest cis-regulatory features discovered in animal genomes—explains how *patched/Ptch1* can drive dramatic adaptations in animal morphology while maintaining its essential core function. It may also suggest a general model for the evolutionary flexibility of conserved regulators and pathways.

## Introduction

Like other major developmental signaling pathways, the Hedgehog (Hh) pathway is broadly conserved among bilaterians, from its basic signal transduction mechanism to the DNA-binding specificity of its effector molecules, the zinc finger transcription factors of the Gli family (*Briscoe and Thérond, 2013*; *Hallikas et al., 2006*). Hh signaling is used for cell-cell communication in many contexts during development, and is maintained into adulthood to control tissue homeostasis. Disruption of signaling has been directly linked to several human cancers such as medulloblastoma and basal cell carcinoma in addition to developmental disorders including spina bifida, exencephaly and cleft lip/palate (*Barakat et al., 2010*; *Scales and de Sauvage, 2009*; *Teglund and Toftgård, 2010*).

The Hh pathway itself is highly pleiotropic, regulating many different cell fate decisions in different cellular contexts, but the vast majority of the direct transcriptional targets of Hh/Gli respond to signaling in a strictly limited, tissue- and stage-specific pattern (*Barolo and Posakony, 2002*). A very small number of target genes, such as *patched* in flies and *Ptch1, Gli1,* and perhaps *Hhip1* in vertebrates, all of which encode components or modifiers of the Hh pathway itself, seem to respond to Hh signaling in a universal, constitutive manner (or nearly so), to the extent that the expression of these target genes is sometimes considered diagnostic for the presence of Hh signaling (*Epstein, 2008*; *Shahi et al., 2015*). Although the regulation of these 'universal' target genes, which respond to Hh/Gli in most or all signaling contexts, has been studied to some extent—for example, an enhancer of *Drosophila patched* was the first direct Hh/Gli target discovered in any organism (*Alexandre et al., 1996*)—the basis of a constitutive response to Hedgehog/Gli, or to any pleiotropic signaling pathway, is not understood. One could imagine a simple 'master' response element that is universally activated by a given signal in all developmental and adult stem-niche contexts where that signal is present. But while efforts have been made to synthesize such generic response elements by multimerizing binding sites, with varying degrees of success (*Barolo, 2006*), so far no such element has been found in nature.

Here we examine the regulation of the *patched* gene, a universal transcriptional target of Hh/Gli signaling in *Drosophila*, and its mouse ortholog *Ptch1*, which has an equally universal response to signaling. The Hh receptor Patched (Ptc) normally inhibits the function of Smoothened, blocking signal transduction and favoring the production of the repressor isoform of the Ci/Gli transcription factor (Gli$^R$). This inhibition is relieved upon binding of Hh ligand to Ptc, allowing the activator isoform (Gli$^A$) to accumulate. Ptc is expressed broadly, keeping the Hh pathway silent in the absence of ligand, but it is also directly transcriptionally activated by Gli in all Hh-regulated tissues, where it moderates signal levels (*Figure 1A*). This constitutive negative feedback circuit, which is conserved from flies to mammals and is essential for normal development, has been extensively studied in *Drosophila*, especially in the contexts of the developing wing and embryonic ectoderm (*Alexandre et al., 1996*; *Briscoe and Thérond, 2013*; *Chen and Struhl, 1996*; *Milenkovic et al., 1999*). A previously characterized promoter-proximal enhancer of *Drosophila patched* (*Alexandre et al., 1996*), referred to here as *ptc$^{prox}$* (*Figure 1B*), contains three optimal consensus Gli binding sites which are required to activate expression in Hh-responsive wing cells (*Figure 1C–E*), but was not examined in other developmental contexts.

Since *patched* is activated in all Hh-responsive tissues, we wondered whether this promoter-proximal element, with its cluster of optimal Gli sites, is capable of reproducing the universal Hh response pattern of its parent gene. Our previous work has demonstrated that high-affinity Gli motifs identical to those in *ptc$^{prox}$* (GACCACCCA) can repress enhancer activity in embryos, even in Hh-responding cells (*Ramos and Barolo, 2013*; *White et al., 2012*). These results presented a potential paradox: the same Gli motifs that have been shown to repress transcription in embryonic stripes are also found in an enhancer of *patched*, a gene which is directly activated by Hh/Gli in those same cells. Two explanations that could reconcile these results were (1) that optimal Gli motifs can either activate or repress transcription in the same cells, depending on their *cis*-context, or (2) that the *ptc$^{prox}$* element is not a universally Hh/Gli-responsive enhancer. To distinguish between these possibilities, we examined the *ptc$^{prox}$* enhancer in the embryonic ectoderm, and found that it fails to respond to Hh/Gli in this context (*Figure 1F,G*). This is also consistent with reports that a larger 3.2 kb fragment including *ptc$^{prox}$* is also insufficient for a complete embryonic response, but larger fragments that include extensive 5' sequences are sufficient to drive *ptc*-like embryo patterns (*Forbes et al., 1993*; *Millard and Martin, 2008*). Because the promoter-adjacent cluster of conserved, optimal Gli motifs produces a tissue-restricted expression pattern, it is not sufficient to explain the universal Hh response of the *patched* gene (*Figure 1C–G*), suggesting that the control region for embryonic stripe expression is located elsewhere in the locus. We then set out to identify the enhancer module responsible for *ptc* expression in embryonic ectodermal stripes and to characterize its Hh/Gli response, hypothesizing that it may depend on lower-affinity Gli inputs than the *ptc$^{prox}$* element.

Our findings, described below, show that the seemingly simple constitutive Hh response of *patched* is controlled by an unexpectedly complex multi-modular system of enhancers spread across the *ptc* locus. The mouse *Ptch1* locus seems to be regulated by a very similar overall structure, making this an uncommonly ancient *cis*-regulatory strategy in the animal genome, perhaps as old as the Hh-Gli-Ptc negative feedback circuit itself. We propose that this regulatory structure, in which many

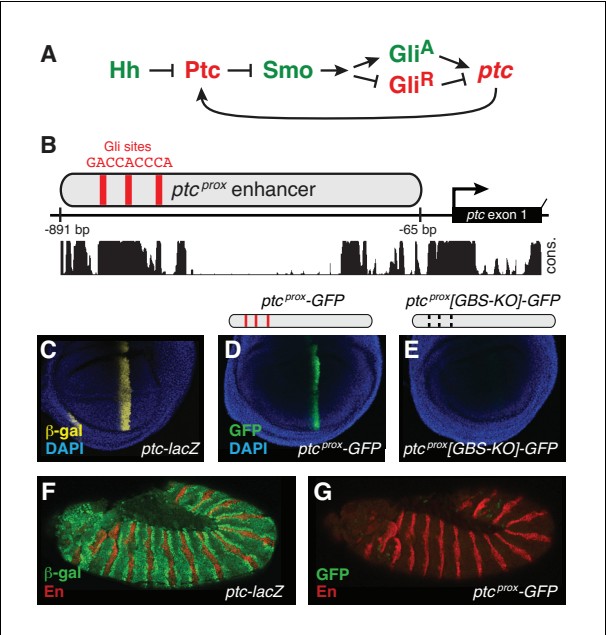

**Figure 1.** The promoter-proximal Hedgehog/Gli-responsive enhancer of *patched* is not sufficient to respond to Hh signaling in the embryo. (**A**) Summary of the conserved Hh-Gli-Ptc pathway feedback loop. (**B**) Promoter-proximal region of *D. melanogaster* patched (*ptc^prox^*). Optimal Ci/Gli binding sites are shown as red bars; sequence conservation among *Drosophila* species is indicated by the histogram at bottom. (**C**) Wing imaginal disc from a *ptc-lacZ* enhancer-trap larva, showing the ptc wing expression pattern. (**D–E**) Larval wing discs carrying *ptc^prox^-GFP* reporter transgenes with the minimal hsp70 promoter and either intact (**D**) or mutated (**E**) Gli sites. Sequences can be found in ***Supplementary file 1***. (**F**) Stage 12 *ptc-lacZ* embryo. (**G**) The *ptc^prox^* enhancer is unresponsive to Hh in the embryonic ectoderm.

context-specific Hh-responsive enhancers together produce a 'simple' constitutive response pattern, can explain why *patched* is an important locus for morphological divergence: it allows Hh signaling levels to be adjusted independently in each tissue and stage, without disturbing pathway activity in other developmental and stem cell contexts.

## Results

### The universal Hh response of *patched* is mediated by a large array of stage- and tissue-specific enhancers

For this study, we created a nomenclature of sequence fragments by dividing the 5' intergenic sequences into regions A through Z (with A containing the *ptc* transcription start site) and the first intron into regions 1A through 1I, and then naming each sequence fragment after the 5'- and 3'-most regions included in that fragment (***Figure 2A***): for example, the *ptc^prox^* enhancer, which spans fragments D, C, and B, will be referred to in the rest of this report as module DB. To reduce bias in our search and to account for the possibility of multiple embryonic enhancers, we independently tested overlapping fragments of the large 5' intergenic region and the large first intron of *ptc* as single-copy, targeted-insertion reporter transgenes in which GFP was driven by the *ptc* promoter (fragment BA) (***Figure 2A***). We focused on the 5' intergenic and first-intron sequences of *ptc* in this functional analysis because the other four introns of *ptc* are short (<350 bp) and poorly conserved and the 3' neighboring gene is located less than 700 bp downstream of *ptc*.

The BA promoter fragment alone drives broad, low-level gene expression in most or all tissues—suggesting that the broad, non-Hh-regulated aspect of *patched* expression may be controlled by this region—but it is not able, on its own, to drive Hh-responsive patterns in any tissue (***Figure 2J, J'*** and ***5C,D***). FAIRE-seq and DNAse-seq datasets (***McKay and Lieb, 2013***; ***Thomas et al., 2011***) show that chromatin states in non-coding regions of the *patched* locus are highly dynamic in the

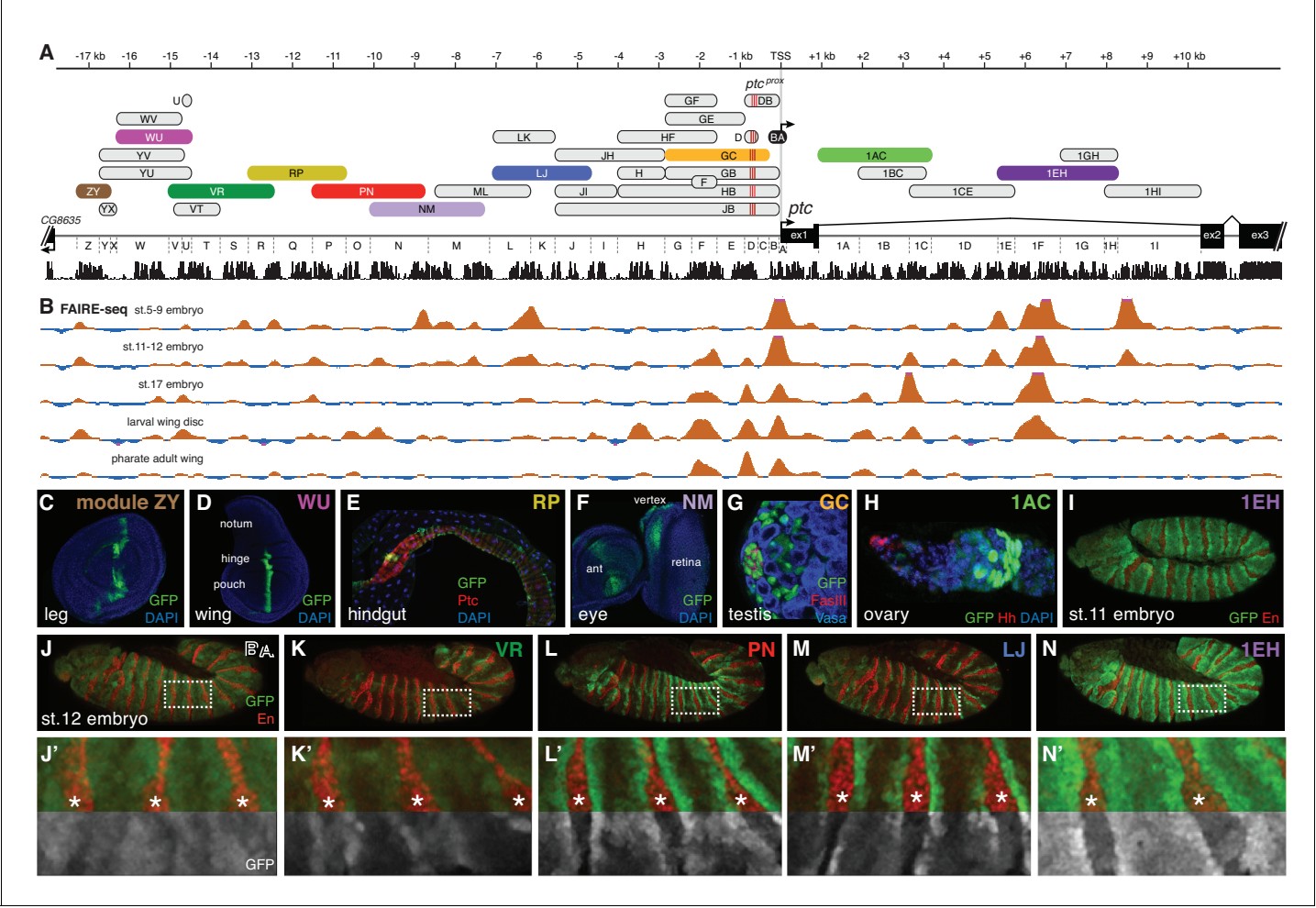

**Figure 2.** The universal Hedgehog response of *patched* is mediated by a large array of tissue- and stage-specific enhancers. (A) The *ptc* genomic locus. DNA regions tested for in vivo enhancer activity are shown as oblong shapes; a sequence conservation histogram is shown below the exon map. (B) In vivo DNA accessibility at the *ptc* locus, as determined by FAIRE-seq across various developmental stages and tissues (*McKay and Lieb, 2013*); DNAse-seq data are shown in *Figure 2—figure supplement 1*. (C–I) Selected larval, adult, and embryonic tissues from GFP transgenic animals. Enhancer names are as in (A). (J) Activity of the *ptc*BA promoter region in a stage 12 embryo, driving broad, low expression in the anterior (En-negative) compartment of each segment. (K–N) *ptc*-like ectodermal stripes driven by four separate *ptc* enhancers in *cis* to the ptcBA promoter. (J'–N') Enlarged views of dashed boxes in (J–N); asterisks show stripes of En-positive, Hh-producing cells in the posterior compartment of each segment. GFP signals are isolated (greyscale) in the lower half of K'–N' to show the lack of stripe enhancer activity in the posterior (En-positive) compartment. Annotated enhancer sequences and PCR primers are provided in *Supplementary file 1* and *Figure 2—source data 1*.

The following source data and figure supplement are available for figure 2:

**Source data 1.** PCR primers used to amplify *Drosophila* genomic sequences.

**Figure supplement 1.** Individual *patched* enhancers exhibit tissue-restricted responses to Hedgehog signaling.

*Drosophila* embryo and larval wing (*Figure 2B*; *Figure 2—figure supplement 1*). Furthermore, the promoter-proximal consensus Gli sites, which are essential for *ptc*[prox] activity in the wing (*Figure 1D–E*), are in regions of mostly closed chromatin in the embryo, consistent with reporter data suggesting that Hh/Gli regulates *patched* via different enhancers in the embryo vs. wing. Every tested fragment in this 27 kb region drives expression in *ptc*-positive cells within at least one Hh-patterned tissue (*e.g.*, *Figure 2C–I*; see also data from *Jenett et al., 2012*; *Kvon et al., 2014*). These tissues represent cell types from nearly every stage of the *Drosophila* life cycle, from multiple embryonic tissues types to larval imaginal discs (*Figure 2C,D,F*) to adult tissues such as the gut and stem

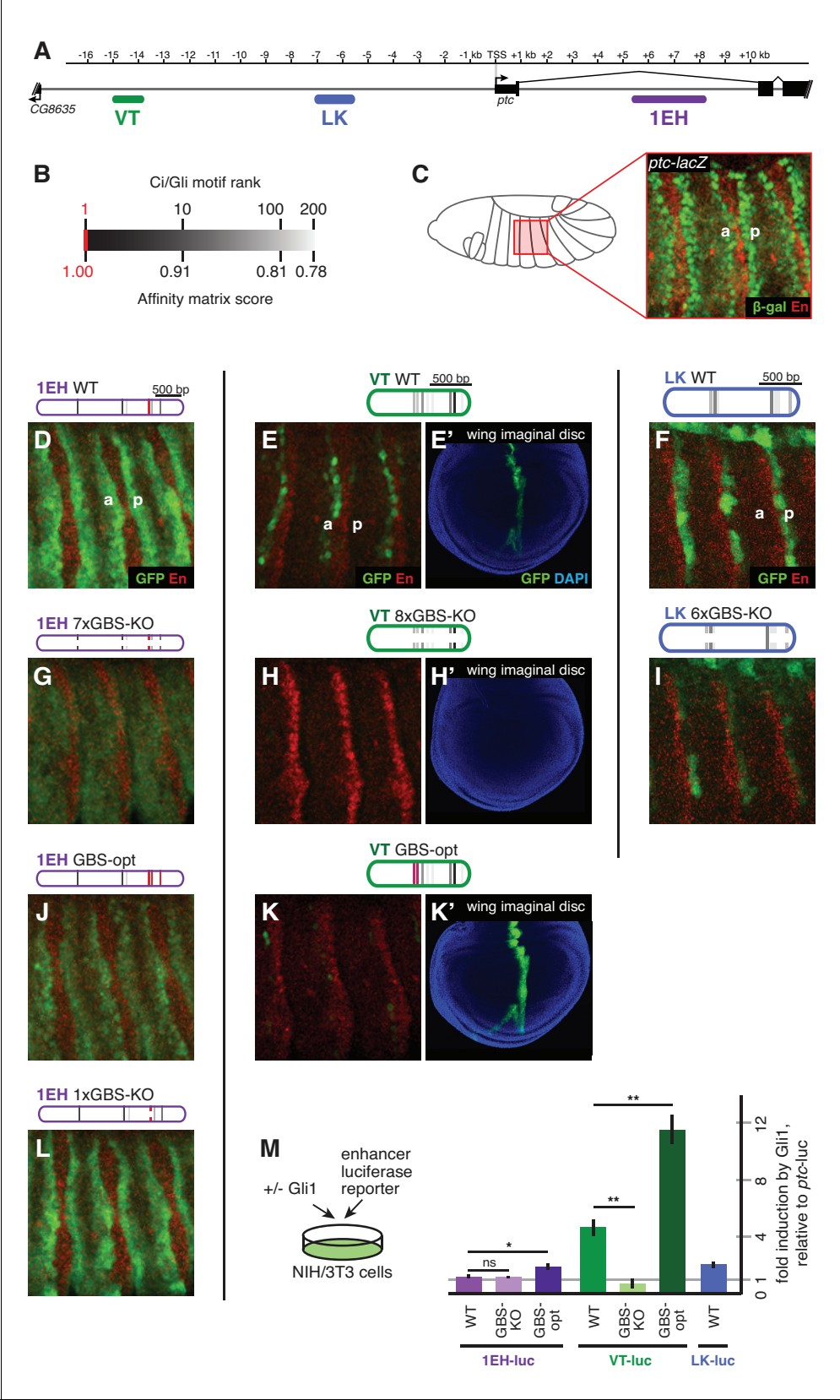

**Figure 3.** Embryonic ectoderm enhancers of *patched* require suboptimal Gli sites to respond to Hh signaling. (A) Map of the *ptc* locus showing enhancers tested in this figure. (B) Scale of predicted relative binding affinity for Ci, used to label predicted Ci/Gli motifs in enhancer diagrams; the

Figure 3 continued

optimal motif (in red) is GACCACCCA. (C) Cartoon depiction of the embryo with the representative region in this figure highlighted in red, *left*. Stage 13 embryo carrying the *ptc-lacZ* enhancer-trap, *right*. (D,H,K) Expression of wild-type (WT) 1EH, VT and LK enhancers in embryos (wing imaginal disc displayed in H'). Full embryos for all panels can be seen in *Figure 3—figure supplement 1*. (E,I,L) Gli binding site knockout (GBS-KO) versions of those enhancers, (wing imaginal disc displayed in I'). In addition, enhancer LK also responds to Hh signaling via overexpression (*Figure 3—figure supplement 2*) (F,J) Optimization of two non-consensus Gli mofits (GBS-opt) in enhancers 1EH and VT (wing imaginal disc displayed in J') (G) The single optimal consensus motif is mutated in the 1EH module (1xGBS-KO). The relative affinity of GBS were assessed by competitive EMSA analysis (*Figure 3—figure supplement 3*, *Figure 3—source data 1*). All enhancer and Gli sites can be found in *Supplementary file 1*. (M) Gli1 responsiveness of embryonic enhancers in NIH/3T3 cells, error bars indicate s.d. Student's t-test; *p<0.05; **p<0.005.

The following source data and figure supplements are available for figure 3:

**Source data 1.** EMSA oligonucleotides used in GBS competition binding assays.

**Figure supplement 1.** Embryonic ectoderm enhancers of *patched* require nonconsensus, suboptimal Gli sites to respond to Hh signaling.

**Figure supplement 2.** *patched* enhancer LK responds to Hh signaling in the embryonic ectoderm.

**Figure supplement 3.** Ci binds to non-consensus Gli motifs in *patched* embryonic ectoderm enhancers in vitro.

cell/niche systems in testes and ovaries (*Figure 2E,G,H*). Hh signaling participates in the development and maintenance of all of these tissues and cell types (*Chen and Struhl, 1996*; *Michel et al., 2012*; *Sahai-Hernandez and Nystul, 2013*; *Takashima and Murakami, 2001*; *Zhang and Kalderon, 2001*). In many tissues, such as the embryonic ectoderm, multiple *ptc* modules are active. However, no single enhancer recapitulates the complete *ptc* expression pattern (*Figure 2C–I*; *Figure 2—figure supplement 1*). Together, these findings are consistent with DamID data indicating that Ci, the *Drosophila* Gli, occupies the entire *ptc* locus in embryos (*Biehs et al., 2010*). Distributed, multi-modular enhancer arrangements have been observed at other developmental gene loci (*Barolo, 2012*), but *ptc* appears to be an unusually complex *cis*-regulatory locus—an idea supported by unusually dynamic chromatin structure (*Figure 2B*; *Figure 2—figure supplement 1*) (*Thomas et al., 2011*).

## Bypassing enhancer-promoter specificity reveals diverse sub-patterns among segmentation stripe enhancers of *patched*

We find that, as in other tissues, the control of *patched* expression in the embryonic ectoderm is distributed among multiple stripe enhancers across the locus. A small promoter element, fragment BA, which does not include the cluster of three optimal Gli sites (*Figure 2A*), drives broad, low-level, non-Hh-regulated expression (*Figure 2J,J'*), while several separable embryonic enhancers, when examined in vivo as GFP reporters containing the BA promoter element, drive *patched*-like segment-polarity stripe patterns (*Figure 2K–N*). In order to distinguish the information encoded in these embryo stripe enhancers from that provided by the endogenous *ptc* promoter, we also tested several *ptc* stripe enhancers with a minimal heterologous *Hsp70* TATA/Inr-containing promoter ('hspmin') driving GFP. When paired with the hspmin promoter, three *ptc* enhancers drive *ptc*-like responses in the embryonic ectoderm (*Figure 3A*; *Figure 3—figure supplement 1*). The intronic element 1EH is activated in stripes to the anterior and posterior of each segmental stripe of Hh-expressing cells, much like the *ptc* gene's expression pattern in this tissue, though with broader stripes (*Figure 3C–D*). The distal 5' enhancer VT is active in a subset of *ptc*-positive cells on the anterior side of Hh-producing stripes in each segment (*Figure 3H*), while the more proximal 5' enhancer LK drives expression in *ptc*+ cells to the posterior of each Hh stripe (*Figure 3K*). Conditional overexpression of the Hh ligand augmented the posterior-stripe expression of enhancer LK, but did not affect the context specificity of its expression pattern (*Figure 3—figure supplement 2*).

## Gli binding site affinity informs tissue specificity

We analyzed the sequence of the embryonic stripe enhancers described above to identify putative binding sites for the zinc finger transcription factor Cubitus interruptus (Ci), the Drosophila Gli, using an in vitro binding dataset (*Hallikas et al., 2006*) which corresponds well with Gli binding preferences in vivo (*e.g., Peterson et al., 2012*) (see Materials and methods). These Ci/Gli motifs were then

altered by overlap extension PCR to abolish Ci binding (see *Figure 3—figure supplement 3* for in vitro binding assays). Mutating Ci/Gli motifs in each embryonic stripe enhancer of *ptc* severely reduced its Hh responsiveness in vivo (*Figure 3E*), demonstrating that low- to moderate-affinity Gli motifs—which account for most or all of the Ci/Gli sites in these modules (*Figure 3—figure supplement 3*)—are required for strong activation in the embryonic ectoderm. Enhancer 1EH contains one optimal GBS (GACCACCCA), but targeted mutation of this motif did not have a strong effect on enhancer activity (*Figure 3G*, compare to *Figure 3D*).

To test whether the low affinity of Gli sites in these enhancers contributed to transcriptional activation, we converted two of the low-affinity motifs in both the 1EH module and the VT module to optimal GACCACCCA motifs. Because this two-site optimization increased Ci binding to those sites in vitro (*Figure 3—figure supplement 3*), and augmented the Gli1-responsiveness of the enhancer modules in mammalian cells (*Figure 3M*), we conclude that the sequence alteration improves Ci/Gli binding as predicted. In vivo, however, the Gli-optimized 1EH and VT enhancers are significantly less active in the embryonic ectoderm than their wild-type counterparts (*Figure 3F* compare to *Figure 3D*). Though perhaps counterintuitive, this finding is consistent with our group's previous reports that optimizing Ci/Gli motifs in Hh-regulated embryonic ectodermal enhancers of the *wingless* and *stripe* genes causes a decrease in enhancer activation in vivo (*White et al., 2012*; *Ramos and Barolo, 2013*). Interestingly, this effect appears to be tissue-/stage-dependent: optimizing two Gli motifs in the VT module decreases its activity in the embryonic ectoderm, as described above, but the same alteration conversely causes an increase of enhancer activity in the wing disc (*Figure 3J'*), where VT is normally weakly active (*Figure 3H'*). These results suggest that there is no level of Gli occupancy that allows optimal activity of this *ptc* enhancer in both the embryonic segments and the wing disc. Thus, nonconsensus Gli motifs can influence both the strength and the stage/tissue specificity of transcriptional responses to Hedgehog signaling.

## Multiple wing disc enhancers of *patched* depend on Gli motifs of varying affinities to synergistically activate gene expression in response to Hh

The *ptc* response to Hh/Gli in the wing, as in the embryonic ectoderm, is distributed across many regions of the locus; in fact, most of the enhancers we tested in vivo had some activity in wing imaginal discs, though not all drove a strong or strictly *ptc*-like expression pattern. We examined a few of these enhancers (*Figure 4A*)—all of which correspond to regions of accessible chromatin in wing discs (*Figure 2B*; *McKay and Lieb, 2013*)—in greater detail with respect to their direct Gli inputs, using GFP reporters with a minimal TATA+Inr 'hspmin' promoter. All of these wing stripe enhancers depend on Gli motifs for full activity in the wing imaginal disc (*Figure 4B*). However, the number and quality of Gli motifs in these modules bears no obvious relationship to the strength of enhancer activity in Hh-responding cells. For example, the previously characterized wing enhancer *ptc*<sup>prox</sup> (referred to here as module DB), with its cluster of three well-conserved optimal sites, is a weaker wing enhancer than several other regions of the *ptc* locus with fewer and/or lower-affinity Gli motifs (*Figure 4B–D*; *Figure 4—figure supplement 1*). Presumably this reflects a requirement for other transcription factor inputs into these modules, but it also demonstrates that binding site quality and number are not always accurate predictors of transcriptional responsiveness in vivo.

When these five wing enhancers (paired with the same hspmin promoter) were tested for Gli responsiveness in Gli1-expressing NIH/3T3 cells as luciferase reporters, we found that their levels of activation were not well correlated with either their expression levels in wing discs or the number or quality of Gli sites (*Figure 4E*, compare to *Figure 4C*). However, the relative change in expression upon mutation of the Gli motifs, both in vitro and in vivo, was usually greater in modules with higher-quality Gli sites such as DB and ZY, compared to modules with weaker sites such as HF and YU (*Figure 4D*). Taking these results altogether, we conclude that, while Gli motif quality plays a role in Hh/Gli responsiveness (positively or negatively, depending on the developmental context), it is not in itself a good indicator of which enhancers will be the most strongly active in a given tissue. Note that of the five wing enhancers examined here, the two with the highest expression levels in wings (HF and YU) have relatively few and/or low-quality Gli motifs, which nevertheless make significant contributions to their activity.

We were intrigued to learn from the above analysis that the canonical *patched* wing enhancer *ptc*<sup>prox</sup>/DB, with its cluster of optimal Gli motifs, drives relatively weak expression in vivo (though it is

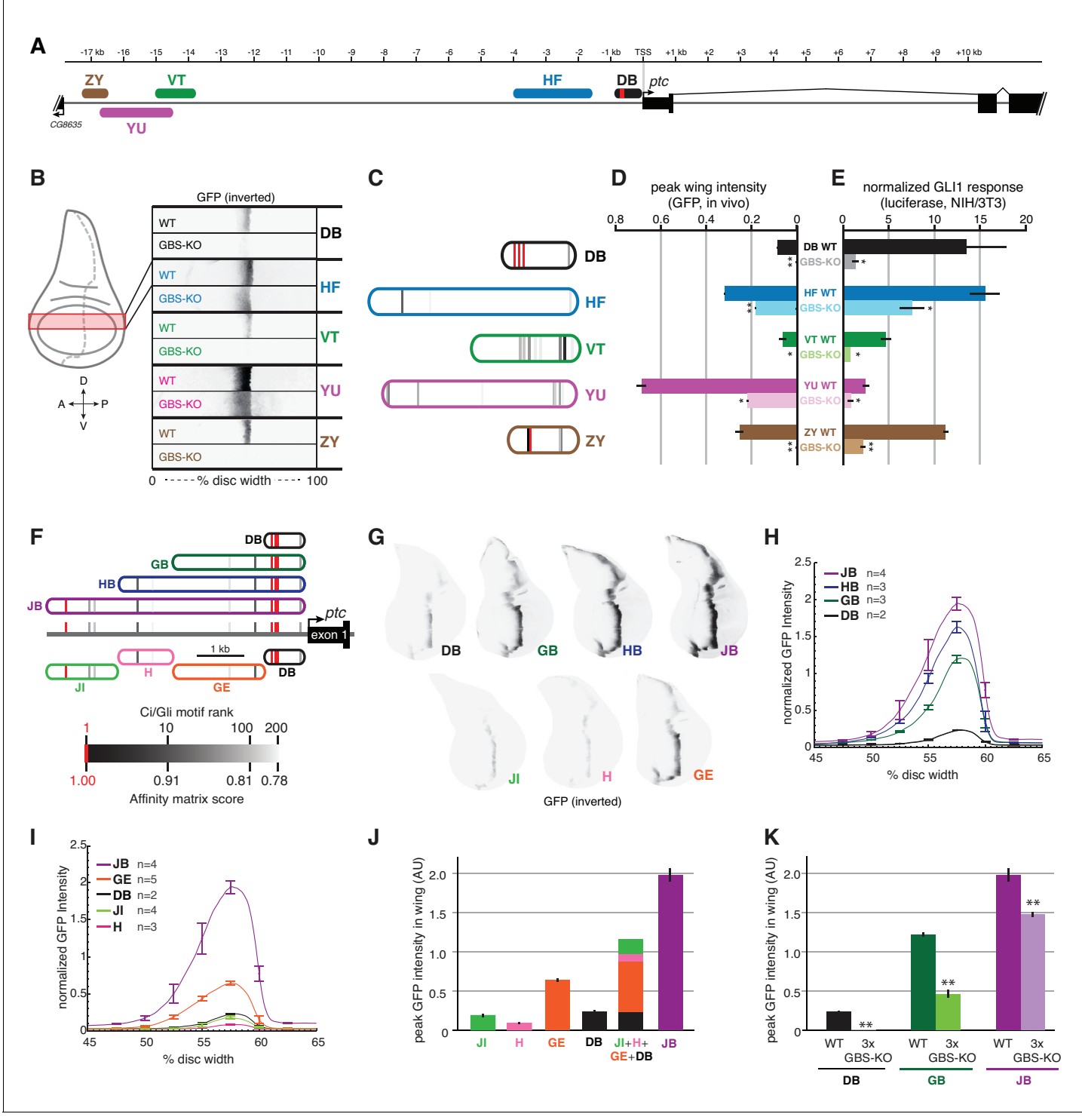

**Figure 4.** Wing enhancers of *patched* synergize to respond to Hh/Gli, largely via low-affinity GBS. (A) Map of the *ptc* locus showing enhancers tested in this figure. (B) Activity of WT and GBS-KO versions of five *ptc* enhancers across the wing imaginal disc. Color-inverted grayscale images of disc segments spanning the anterior-posterior axis; GFP fluorescence appears black. The relative affinities of GBS were assessed by competitive EMSA analysis (*Figure 4—figure supplement 1*). (C) Diagrams of five wing enhancers and their GBS; all enhancers are color-coded as in (A). (D) Peak wing disc GFP intensity driven by WT and GBS-KO wing enhancers. (E) Gli1 responsiveness of the same enhancers in NIH/3T3 cells, error bars indicate s.d. Student's t-test; *p<0.05; **p<0.005. (F) Tested enhancers ≤6 kb upstream of the *patched* promoter are mapped to the *ptc* locus with best predicted high- and low-affinity GBS in red and gray, see GBS affinity color scale, bottom. (G) Promoter-proximal enhancer activity in wing imaginal discs of transgenic larvae (inverted grayscale represents GFP expression). (H,I) Quantitation of GFP enhancer activity across the wing disc. (J–K) Quantitation of
*Figure 4 continued on next page*

*Figure 4 continued*

peak GFP fluorescence in transgenic wing discs carrying enhancer-GFP reporters. All enhancer sequences and Gli binding sites can be found in **Supplementary file 1**.

The following figure supplement is available for figure 4:

**Figure supplement 1.** Gli binds to low affinity motifs in wing enhancers.

strongly responsive to Hh/Gli in vitro). This result led us to hypothesize that the high-affinity Gli sites in the DB module may require additional flanking sequence in order to exert a strong influence on gene expression in vivo. We first addressed this by testing a nested set of fragments from the -5.5 kb region upstream of *ptc,* all containing the 3 proximal optimal GBS (*Figure 4F*, *top).* All of the fragments (DB, GB, HB, and JB, in order of increasing size) drove *ptc*-like stripes in the wing, but while the anterior-posterior positions of these stripes were the same, the levels and dorsal-ventral extents of gene expression increased greatly with the addition of 5' flanking sequence (*Figure 4G*). 5' sequences containing relatively few and weak Gli motifs provided significant boosts to gene expression in the wing: for example, compare GB to DB, and HB to GB (*Figure 4G*). We next examined the direct contribution of the three optimal Gli sites (located within region D) to the in vivo activity of the nested fragments DB, GB, and JB. In all cases, mutating these three Gli sites (while leaving any other Gli motifs intact) significantly reduced peak gene expression in the wing stripe (*Figure 4K*) (Student's t-test; p<0.0015). However, the relative contribution of those three sites, measured as reduction of expression of a 3xGli-mutant enhancer compared to wild-type, decreased as more *cis*-regulatory context was included—from a 99.9% reduction in module DB, to a 62% reduction in module GB, to a 26% reduction in module JB (*Figure 4K*). When taken out of their native sequence context and tested as a synthetic reporter, three optimal Ci/Gli sites are not capable of activating gene expression in the wing, though they can strongly synergize with other activator binding sites (*Ramos and Barolo, 2013*).

By breaking down module JB into smaller, non-overlapping sub-fragments (JI, H, GE, and DB; see *Figure 4F*), we were able to test the individual contributions of these regions, with very different predicted Gli binding patterns, to gene activation. Note that of these fragments, only module DB contains the cluster of three optimal Gli motifs. All of these modules were sufficient to drive expression in the wing disc, and as with the more distal *patched* wing enhancers examined above, little or no correlation between predicted Gli affinity and in vivo expression levels was observed (*Figure 4G*). In general, transcriptional synergy among sub-fragments was observed in vivo, such that the activity of a large fragment (JB, HB, GB) was greater than the sum of the individual activities of its constituent sub-fragments (*Figure 4H–J*).

The above findings suggest that, even in the wing, the promoter-proximal cluster of optimal Gli sites makes a relatively minor (though significant) contribution to the Hh/Gli response of *patched*, and that inputs from many additional *cis*-regulatory regions, some of which contain relatively few and/or weak Gli motifs, are integrated to produce the final pattern of transcriptional activation in this tissue.

## An unusual tissue- or stage-specific Polycomb/Trithorax response element (PRE) alters the output of *patched* enhancers

In addition to the multi-modular complexity of the *ptc* locus, we found evidence of higher-level control of gene expression via specialized enhancer-promoter interactions. Some *ptc* enhancers drove qualitative and quantitatively different in vivo expression patterns when placed in *cis* to a homologous vs. a heterologous promoter (*e.g.,* *Figures 2M,3K*; see also below). In third-instar larvae, the *ptc* promoter and 5' sequences are bound by the Polycomb group (PcG) factors Pleiohomeotic (Pho), Polyhomeotic (Ph), Polycomb (Pc), and Posterior sex combs (Psc), as well as the Trithorax group (TrxG) factor Trl/GAF (*Oh et al., 2013*; *Schaaf et al., 2013*), and are marked with the repressive histone modification H3K27me3 (*Figure 5A*). These are all hallmarks of Polycomb/Trithorax response elements (PREs), *cis*-regulatory sequences which mediate epigenetic regulation by PcG/TrxG factors (*Brown and Kassis, 2013*; *Schaaf et al., 2013*; *Bowman et al., 2014*). In embryos, however, the *ptc* promoter lacks PRE signatures: compared to the PRE-containing loci *en/inv, AbdB*,

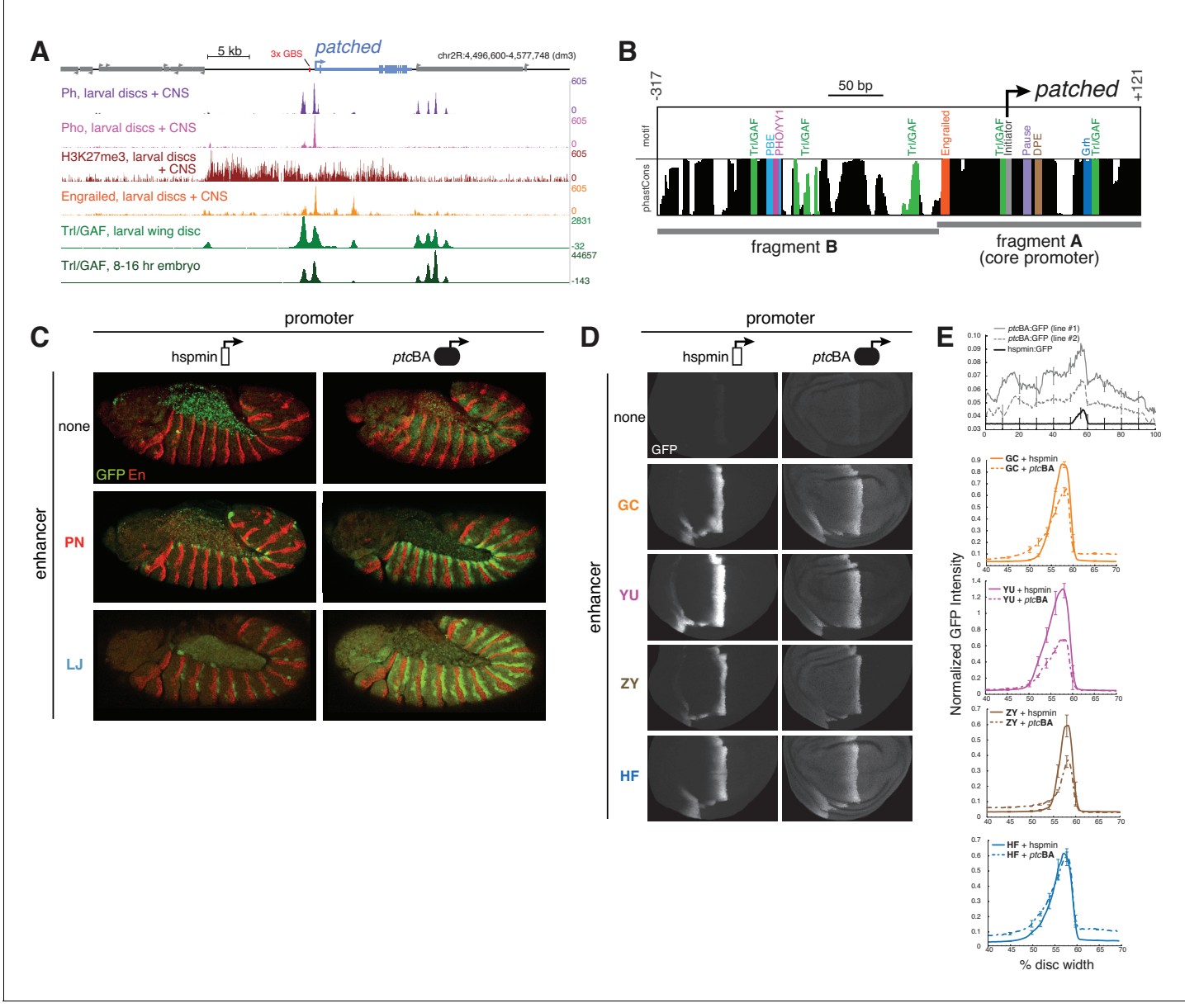

**Figure 5.** Developmentally dynamic Polycomb/Trithorax Response Elements (PREs) near the *patched* promoter modulate enhancer activity differently in the embryo and developing wing. (**A**) ChIP-seq data from larval tissues showing binding of PcG proteins, the H3K27me3 histone mark, Engrailed, and Trithorax-like(Trl)/GAF in the neighborhood of *ptc*. Embryonic Trl/GAF binding data (*bottom*) are taken from ***Oh et al., 2013***. For comparison with larval PcG/TrxG binding, see embryonic ChIP-seq data in ***Figure 5—figure supplement 1***. (**B**) DNA motif analysis of the *ptc*BA promoter. Additional 5' flanking sequence is analyzed in ***Figure 5—figure supplement 2***. (**C**) Two embryonic ectodermal enhancers of *ptc* (rows) tested on different promoters (columns) in stage 12 transgenic embryos. Related experiments are presented in ***Figure 5—figure supplement 3***. (**D**) Four wing enhancers of *ptc* tested on a minimal TATA+Inr promoter (hspmin) or the native ptc promoter (fragment BA; -317 to +121). (**E**) Quantitation of wing GFP expression across wing imaginal discs carrying the transgenes shown in panel **D**.

The following source data and figure supplements are available for figure 5:

**Source data 1.** Conserved TF motifs identified in *ptc* enhancer sequences.

**Figure supplement 1.** The *patched* promoter region, which is heavily bound by PcG proteins and H3K27me3-marked in larval discs, does not show PRE signatures in the embryo.

**Figure supplement 2.** Conserved PRE-associated transcription factor binding motifs in promoter-proximal sequences of the *patched* gene.

*Figure 5 continued on next page*

*Figure 5 continued*

**Figure supplement 3.** Polycomb/Trithorax Response Elements (PRE) near the patched promoter modulates enhancer activity.

and *Ubx*, the *ptc* promoter region has low PcG occupancy (except for Pho) and low H3K27me3 (*Schuettengruber et al., 2009*) (*Figure 5—figure supplement 1*). A motif analysis of DNA sequences located 5' of the *ptc* promoter using the CIS-BP database (*Weirauch et al., 2014*) identified conserved binding motifs for the PcG protein Pho (ortholog of vertebrate YY1) and the TrxG transcription factor Trl/GAF (*Figure 5B*; *Figure 5—figure supplement 2*), consistent with our tissue-specific ChIP-seq binding data. These motifs are frequently enriched in PREs and often required for their function (*Brown and Kassis, 2013*). We also compared the core promoter motifs present in our two test promoters. The *ptc* core promoter region lacks a TATA box but contains a *Drosophila* Initiator element (Inr) and multiple GAF-binding GAGA motifs, as well as a pause button (PB) and a Downstream Promoter Element (DPE) 3' of the transcription start site (*Figure 5B*; *Figure 5—figure supplement 2*). The GAGA + Inr + [PB and/or DPE] promoter configuration is associated with RNA polymerase II stalling at *Drosophila* promoters (*Hendrix et al., 2008*), and embryonic GRO-seq data identify *ptc* as one of the most heavily promoter-paused genes in the *Drosophila* genome (*Saunders et al., 2013*). The *ptc* promoter is highly dissimilar from the minimal *Hsp70* 'hspmin' promoter fragment used here as an alternative heterologous promoter, which has a TATA + Inr motif configuration (*Emanuel and Gilmour, 1993*).

When paired with the heterologous hspmin promoter, the *ptc* embryonic stripe enhancers PN and LJ activate expression in restricted subsets of the *ptc* segment-polarity stripe pattern (dorsal and posterior *ptc*-positive cells, respectively, within each embryonic segment) (*Figure 5C*, *left*). However, when paired with the native *ptc* promoter fragment BA—which alone is broadly and weakly active (like the *ptc* gene itself) but unresponsive to Hh/Gli (*Figures 2J*,*5J'* , *top*)—these enhancers drive stronger and more complete, *ptc*-like expression patterns in embryos (*Figure 5C*, *right*). Neither the core *ptc* promoter, fragment A, nor the upstream element, fragment B, were sufficient to enhance the Hh/Gli response of module PN (*Figure 5—figure supplement 3A–H*), suggesting that both the *ptc* core promoter, region A, and the promoter-proximal fragment B are required for this enhancement.

We next examined the promoter effects on *ptc* enhancer function in larval imaginal discs—where, in contrast to the embryo, we found strong PcG binding and the Polycomb-produced repressive mark H3K27me3 at and upstream of the promoter (*Figure 5A*; compare with *Figure 5—figure supplement 1*). Three of four tested wing-stripe enhancers (GC, YU, and ZY) showed reduced activity when paired with the *ptc*BA promoter, compared to the heterologous hspmin promoter (*Figure 5D*). A fourth wing enhancer, HF, showed no difference in peak levels of activation, but had a higher baseline expression level across the wing when paired with *ptc*BA (*Figure 5D*).

Taken together, our genomics and functional results suggest that interactions among multiple regions of the *cis*-regulatory apparatus, including distal enhancers, 5' promoter-proximal sequences, and the core promoter are integrated to determine the final pattern and levels of gene expression in each tissue, in a manner that correlates with the presence or absence of PcG/TrxG regulation of a tissue- or stage-specific PRE.

Interestingly, the TrxG factor GAF—which has been associated with transcriptional activation in addition to its role in PRE-mediated repression (*Adkins et al., 2006*)—directly occupies the intronic embryonic stripe enhancer 1EH in the embryo, though to a lesser extent than the *ptc* promoter (*Figure 5A*; *Figure 5—figure supplement 1*), probably via conserved GAGA motifs in that element (*Figure 5—source data 1* and *Supplementary file 2*). This may explain why the 1EH module, the most powerful embryonic stripe enhancer at the *ptc* locus, does not require the heavily GAF-occupied *ptc* promoter for strong activation in the embryo (*Figure 5—figure supplement 3*). GAF has recently been implicated in large-scale changes in gene regulation during the embryo-larva transition (*Blanch et al., 2015*).

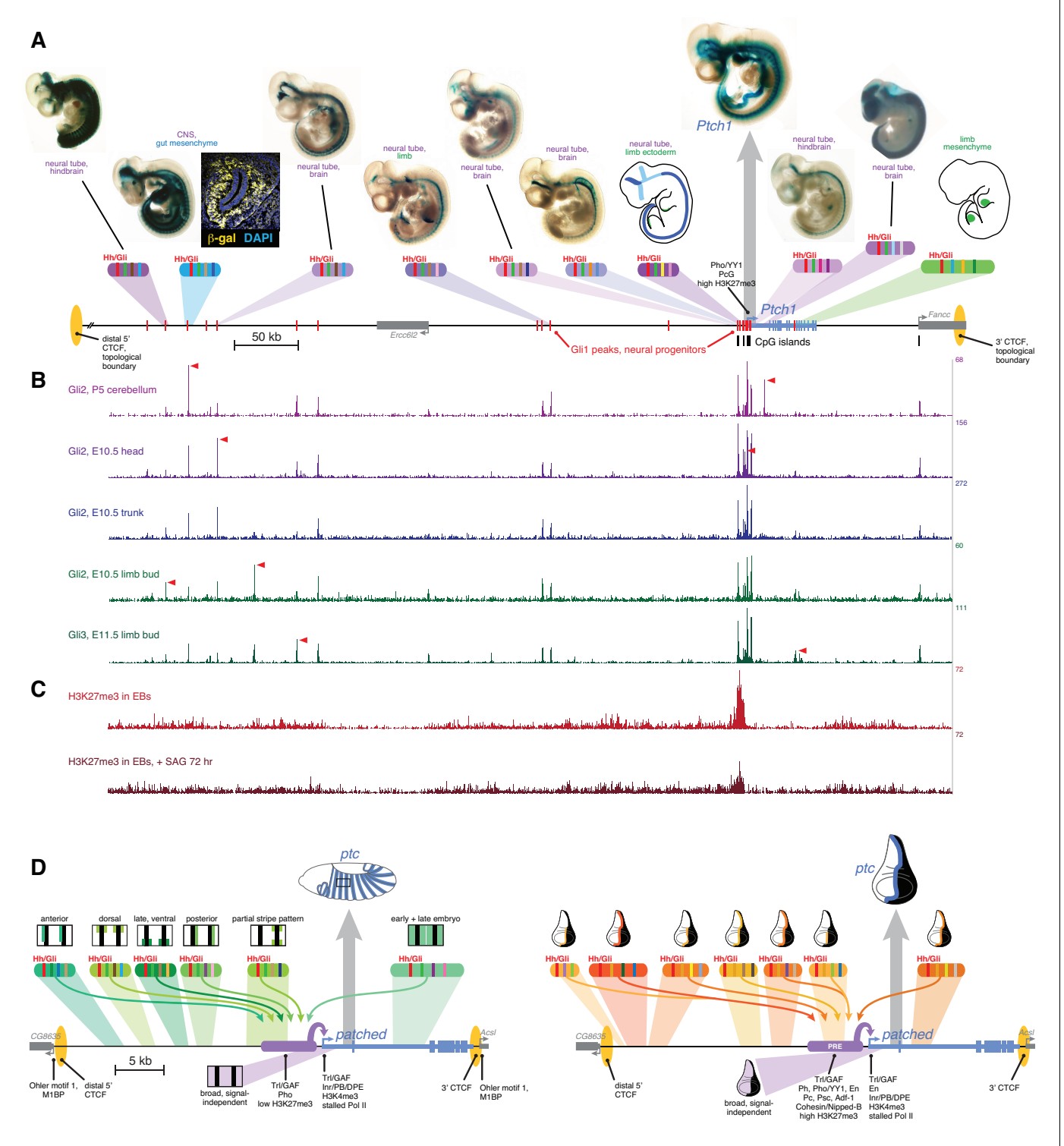

**Figure 6.** *Drosophila* and vertebrate *patched* orthologs share a similar *cis*-regulatory architecture and gene regulatory strategy. (**A**) X-gal-stained E10.5 mouse embryos carrying novel *Ptch1* enhancers as transient-transgenic *lacZ* reporters (*Figure 6—source data 1*) are shown above a map of the mouse *Ptch1* region. Known *Ptch1* enhancers (*Vokes et al., 2007*; *Lopez-Rios et al., 2014*) are shown as cartoons. Red boxes along the gene locus schematic represent Gli1 occupancy in neural progenitors near *Ptch1* (as identified by ChIP-seq; *Peterson et al., 2012*). (**B**) ChIP-seq of Gli2 and Gli3 near *Ptch1*. Red arrowheads highlight tissue-specific Gli binding. (**C**) ChIP-seq shows H3K27me3 is enriched at the proximal promoter but reduced upon pathway stimulation, consistent with relief of Polycomb-mediated repression. (**D**) *Left*, summary of the *ptc* response to Hh in the *Drosophila* embryonic ectoderm. Multiple Hh-responsive enhancers combine Ci/Gli sites (red bands) and tissue-specific inputs (colored bands), generating overlapping

*Figure 6 continued on next page*

*Figure 6 continued*

subsets of the embryonic *ptc* response which are integrated and amplified by the promoter-proximal region, resulting in a complete segment-polarity stripe pattern (top). The promoter-proximal region also generates the basal signal-independent *ptc* expression (bottom, purple). *Right*, summary of the Hh response of *ptc* in the wing disc. A large battery of tissue-specific Gli-responsive enhancers, driving wing stripes of different widths, intensities, and dorso-ventral extents, interact with a Polycomb/Trithorax-bound promoter-proximal region to provide a complete wing stripe pattern (top). The PRE-containing promoter-proximal region also produces signal-independent *ptc* expression in the anterior (bottom, purple). Identification of flanking CTCF insulator sites for both fly and mouse *patched* were determined using ChIP-seq and ChIP-CHIP analyses available from the UCSC genome browser and modENCODE databases (*Figure 6—figure supplement 1*).

The following source data and figure supplement are available for figure 6:

**Source data 1.** Mouse *Ptch1* enhancer coordinates.

**Figure supplement 1.** Topological boundaries and chromatin insulators at mouse *PTCH1* and fly *patched*.

## *Drosophila* and vertebrate *patched* orthologs share a similar cis-regulatory strategy

Direct transcriptional activation of *ptc* by Hh/Gli, for the purpose of negative feedback regulation, is a regulatory circuit that predates the protostome-deuterostome split (*Alexandre et al., 1996*; *Goodrich et al., 1996*). In light of this unusually ancient evolutionary linkage, we asked whether the *cis*-regulation of the mammalian *Ptch1* gene resembles that of its insect ortholog. Two Gli-bound enhancers of mouse *Ptch1* have been reported: a high-affinity promoter-proximal neural tube module (*Vokes et al., 2007*) and a lower-affinity intronic limb bud mesenchyme module (*Lopez-Rios et al., 2014*). Our ChIP-seq analysis and previous studies (*Lopez-Rios et al., 2014*; *Peterson et al., 2012*; *Vokes et al., 2007*) identify many sites of significant Gli binding near *mPtch1* and extending far upstream of the gene (*Figure 6A,B*). We tested seven of these Gli-bound regions as lacZ reporters in transgenic mice. As with enhancers of *Drosophila ptc*, we found that all tested *Ptch1* modules are active in different, overlapping subsets of the *Ptch1* expression pattern: no single module, no matter how strongly bound by Gli, recapitulates the complete Hh/Gli-responsive expression pattern of the parent gene (*Figure 6A*). Binding of Gli transcription factors to numerous sites in the *Ptch1* locus is dynamic across several different developmental contexts, perhaps suggesting that enhancer modules are differentially accessible in a developmentally regulated manner (*Figure 6B*, red arrows). Even taken together, the novel enhancers that we identify here and the previously identified modules do not account for the complete *Ptch1* pattern, suggesting that there are still more yet uncharacterized *Ptch1* enhancers to be found—presumably with relatively lower levels of Gli factor binding, since the highest-occupancy ChIP-seq peaks have now been tested. As in fly *ptc,* chromatin at the mouse *Ptch1* locus is H3K27-trimethylated at the promoter and in upstream noncoding regions (*Figure 6C*, compare with *Figure 5A*), characteristic of PcG regulation (*Bowman et al., 2014*; *Brown and Kassis, 2013*).

Mouse *Ptch1* and fly *patched* share a general gene-expression profile (broad, weak basal expression plus a constitutive response to Hh/Gli), as well as a complex multi-enhancer organization in which the universal response to Hedgehog signaling is not constituted in a master Gli response element, but distributed across many sub-tissue-specific Gli-regulated modules (*Figure 6A,D,E*). The Gli-regulated enhancers discovered to date at both the *Drosophila patched* and vertebrate *Ptch1* loci range from remote 5′ elements, to promoter-proximal modules with optimal Gli motifs, to intronic enhancers. Hi-C topographical domain mapping and ChIP-seq occupancy data for the chromatin insulator/boundary factor CTCF indicate that both *Ptch1* and *patched* are flanked by a nearby 3′ boundary (associated with a closely neighboring downstream gene) and a far more remote 5′ boundary (*Dixon et al., 2012*; *Wang et al., 2012*) (*Figure 6A,D,E*; *Figure 6—figure supplement 1*). In both fly and mouse, all mapped enhancers fall within this topographical domain; they rely on Gli sites across a wide range of affinities; they appear to also require additional tissue-specific transcription factor inputs (*Supplementary file 2*); and they act through a PcG-bound, H3K27me3-marked promoter. This combination of regulatory mechanisms, ranging in scale from the single binding site to the locus as a whole, allows independent tuning of *ptc/Ptch1* expression—and therefore of Hh

signaling levels—in individual tissues, without disrupting overall signaling required for normal development and adult tissue homeostasis (*Figure 6A,D,E*).

## Discussion

This study, which examines the regulation of an essential developmental gene at the locus-wide level, demonstrates for the first time how a constitutive pathway-responsive gene is precisely regulated in diverse developmental contexts. This regulation consists of two major modes of control: First, *ptc* must be broadly active to inhibit Smoothened activity, preventing the inappropriate activation of Hh target genes and subsequent faulty patterning, developmental defects, and disease (*Katoh and Katoh, 2009*; *Barakat et al., 2010*; *Scales and de Sauvage, 2009*; *Teglund and Toftgård, 2010*). Our reporter-based analysis of the locus has identified a promoter-proximal 5' sequence, region B, which in conjunction with the core promoter acts as the likely controller of the basal, signal-independent mode of *ptc* expression.

Second, *ptc* must be able to respond to Hh/Gli signaling in many developmental and stem-niche contexts. We show here that this is not accomplished by a master Hh/Gli response element, but rather by a collection of Gli-regulated enhancers spread across the *ptc* gene locus. These remote, tissue- and sub-tissue-specific enhancers work synergistically and are integrated with additional information from promoter-proximal sequences and the core promoter, responding to Hh signaling differently in each stage and tissue type. The basis for the tissue-specific activity of the numerous enhancers of *ptc* has not been directly examined yet, but in some cases we have identified likely candidate selector inputs by a combination of sequence motif searching and DNA conservation analysis (*Supplementary file 2*; *Figure 6D,E*).

At least in some enhancer modules, we find that the quality/affinity of Gli binding motifs in these enhancers is extremely important for proper patterning, to the extent that optimizing Gli motifs in certain enhancers can cripple enhancer activity. Previous studies from our group and others have shown that low-to-moderate-affinity transcription factor binding sites can be important for limiting the response of a target gene to a pleiotropic, broadly active transcription factor or pathway (*Swanson et al., 2011*; *Crocker et al., 2015*; *Farley et al., 2015*; additional references within). Our results with embryonic stripe enhancers of the *ptc* gene, however, point to a different type of regulatory control by weak binding sites, in which the occupancy of a transcription factor is modulated to an optimal level of activation within a tissue. The fact that optimal activation is achieved via suboptimal Gli binding sites in the *Drosophila* embryo may reflect the fact that Ci/Gli acts both as an activator and as a repressor through the same binding sites. Considering that the area and duration of Hh production in embryonic segments is vastly lower than that in larval/pupal wings, where half of the organ produces Hh protein for most of the developmental period of the animal, it is to be expected that the $Gli^A:Gli^R$ ratio may be significantly lower in Hh-responsive cells in the embryonic segments than in late-larval wing discs. If so, the relatively high concentration of $Gli^R$ in Hh-responding embryonic cells may create selective pressures favoring lower Gli occupancy, as we have argued previously (*Parker et al., 2011*; *White et al., 2012*). Alternatively (or in addition), other mechanisms, such as overlapping binding specificities for non-Gli transcription factors, could play a role in the sub-optimization of Gli motifs observed here.

Our results from transgenic mouse reporter constructs and Gli ChIP-seq experiments suggest that the same large-scale *cis*-regulatory strategy at work in the fly locus is also present in vertebrates—at least to the extent that many enhancers, with widely varying levels of Gli input, act in combination but in a tissue-specific manner to independently adjust *Ptch1* expression levels in each cell type, giving the illusion of a universal response to Hh signaling but in fact representing a complex array of context-limited responses. However, we do not mean to propose that the specific *cis*-regulatory sequences that regulate *patched* and its orthologs are conserved across such large evolutionary distances. Individual *patched* enhancers show no obvious sequence conservation from insects to mammals—unsurprisingly, given that most *ptc*-expressing, Hh/Gli-regulated organs present in modern animals did not exist at the time of the protostome-deuterostome divergence. Rather, we propose that *patched* and its orthologs share an ancient regulatory structure composed of relatively young enhancers. Based on our motif analyses and mutational studies, we propose that *patched* enhancer evolution likely involved both tuning of Gli occupancy and the co-option of tissue-specific inputs to produce new aspects of the *patched* expression pattern over time. This elaboration of new

sub-patterns would not be necessary if *patched/Ptch1* were regulated by a unified 'master' Hedgehog response element; under such a simple regulatory scheme, new domains of *patched* expression would appear automatically whenever Hh signaling were activated in new contexts. By contrast, our analysis suggests that a robust response to Hh/Gli in both embryonic segment-polarity stripes and in larval wing stripes may not be achievable by the same enhancer module, perhaps due to conflicting requirements for Gli occupancy, or perhaps for other reasons whose basis in *cis*-regulatory logic is not yet clear.

The *cis*-regulatory architecture discovered here is also a remarkably ancient example of a direct transcriptional linkage and gene regulatory strategy in animals. Notch regulation of Hes/Hey genes is a circuit of comparable age, but its conservation is limited to the presence of CSL-family binding sites controlling orthologous target genes (*Rebeiz et al., 2012*), with no evidence of a larger shared gene-regulatory structure. Conversely, while the collinear structure of animal Hox gene clusters predates the Bilateria, the upstream regulators of Hox genes do not appear to be shared between protostomes and deuterostomes (*Duboule, 2007*). Our results suggest that the locus-wide multimodular structure by which *patched* responds to Hh/Gli to provide feedback inhibition is an unusually old *cis*-regulatory strategy. This strategy has the potential advantages of transcriptional precision and robustness ascribed to regulation by 'shadow' enhancers (*Frankel et al., 2010*; *Perry et al., 2010*; *Barolo, 2012*; *Wunderlich et al., 2015*), but more importantly it allows for the independent modulation of signaling levels in a stage- and tissue-specific manner, without disturbing pathway activity in other contexts.

Our model has significant implications for the evolution of cell signaling pathways and their target genes and tissues. For example, it helps to explain two reported cases of morphological divergence caused by changes at the vertebrate *Ptch1* locus: modifications to an intronic enhancer of *Ptch1* produced major developmental adaptations in the bovine limb (*Lopez-Rios et al., 2014*), while sequence variation upstream of *Ptch1* is linked to cichlid craniofacial diversification (*Roberts et al., 2011*). In both cases, Hh signaling levels in other developmental contexts (for example, the spinal cord) were apparently unaffected by the adaptation, indicating that the affected enhancer module is stage/tissue-specific. Our conserved structure with flexible enhancers model of *patched* regulation is also consistent with the finding that, while zebrafish *ptch1* resembles its mouse and fly orthologs in having multiple Gli-bound regions, few such sites are conserved among vertebrates (*Peterson et al., 2012*; *Wang et al., 2013*). Thus, while the *cis*-regulatory structure of *patched* appears to have ancient roots, it is also highly flexible, allowing a core component of the Hedgehog pathway to function as an adaptable tissue-specific modifier of pathway activity and as a substrate for morphological evolution.

## Materials and methods

### Transcription factor binding site prediction and ranking

Matrix similarity scores were calculated (*Quandt et al., 1995*) using in vitro Ci/Gli binding data (*Hallikas 2006*). GBS were identified in silico by screening the *ptc* locus for defined motifs using GenePalette (*Rebeiz and Posakony, 2004*). PBEs (Polycomb-core-complex Binding Elements) were defined by Mohd-Sarip et al. (*Mohd-Sarip et al., 2005*).

### DNA sequence alignments

Sequences and multi-species alignments were obtained from the UCSC Genome Browser (genome. ucsc.edu).

### DNA cloning and mutagenesis

Wild-type *ptc* enhancers were amplified by PCR (Roche Expand High Fidelity PCR System) from BAC DNA (CH322-170A-12 or CH322-188E13). PCR primers are provided in *Figure 2—source data 1*. Enhancer constructs were sub-cloned into the pENTR/D-TOPO plasmid (Life Technologies) by TOPO cloning. Enhancers tested with the hspmin promoter, taken from the *D. melanogaster Hsp70* gene, were subsequently cloned into the pHPdesteGFP transgenesis vector via LR Cloning (Life Technologies). Enhancers tested with the endogenous *ptc* promoter were cloned by traditional methods into the pStinger transgenesis vector (*Barolo et al., 2000*). Targeted GBS mutations were created by

overlap-extension PCR (*Swanson et al., 2010*). Promoter analysis described in *Figure 5* and *Figure 5—figure supplement 3* was done by replacing the minimal hsp70 promoter contained in pHPdeste with the designated *ptc* promoter region (see *Figure 2—source data 1* for sequences and restriction sites used). All enhancers and promoters were screened by restriction digest and sequencing.

## *Drosophila* transgenesis

P-element transformation was performed as previously described (*Swanson et al., 2010*) in the $w^{1118}$ strain. Site-directed transformation by embryo injection was performed as described by *Bischof et al., 2007*, with reporter transgenes integrated into a ΦC31 landing site at genomic position 86Fb.

## Immunohistochemistry and confocal microscopy

*Drosophila* embryos and third-instar imaginal discs were fixed and stained using standard methods (*Parker et al., 2011*; *Ramos and Barolo, 2013*; *White et al., 2012*). Adult testes and ovaries were dissected between 0–2 days after adult hatching, fixed in 4% paraformaldehyde, washed in PBS with 0.1% TritonX, and subjected to antibody staining. Third-instar larval gut was dissected and fixed in the same matter as testes and ovaries. Primary antibodies used included rabbit anti-EGFP (Invitrogen 1:100), mouse anti-beta-galactosidase (Developmental Studies Hybridoma Bank 40-1a, 1:200), mouse anti-Engrailed (Developmental Studies Hybridoma Bank 4D9, 1:50), mouse anti-Wingless (Developmental Studies Hybridoma Bank 4D4, 1:50). In *Drosophila* embryos, EGFP antibodies were used to visualize reporter expression; in imaginal discs, native GFP fluorescence was imaged directly. Antibodies obtained from the Developmental Studies Hybridoma Bank were developed under the auspices of the National Institute of Child Health and Human Development and maintained by the Department of Biological Sciences, The University of Iowa (Iowa City, IA). AlexaFluor488, AlexaFluor555, and AlexaFluo568 conjugates with secondary antibodies from Invitrogen were used at 1:2000 dilutions. DAPI was included in the Prolong Gold antifade mountant (Life technologies). Confocal images were captured on an Olympus FluoView 500 Laser Scanning Confocal Microscope mounted on an Olympus IX-71 inverted microscope, and on a Nikon A1 confocal microscope. Samples to be directly compared were fixed, prepared, and imaged under identical confocal microscopy conditions and settings. Quantitative GFP expression data from imaginal discs were collected from confocal images: all GFP-expressing fly stocks were crossed to the same reference line, *dpp*D-Ci$^{ptc}$, which drives DsRed expression in wing discs, as an internal normalization reference (*Parker et al., 2011*). DsRed and GFP were imaged for each disc, and GFP levels were normalized to peak DsRed fluorescence (*Parker et al., 2011*). Normalized GFP fluorescence data across wing discs (*Figures 3,5*) were graphed in MATLAB (E. Ortiz-Soto, A.I.R. and S.B., manuscript in preparation). Two-tailed t tests for two samples with unequal variances were used to compare peak wing expression levels of GFP reporters.

## Cell culture assay

NIH/3T3 cells were cultured at 37°C, 5% $CO_2$, 95% humidity in Dulbecco's modified eagle medium (DMEM; Gibco, cat. #11965–092) containing 10% bovine calf serum (ATCC; cat. #30–2030) and penicillin/streptomycin/glutamine (Gibco, cat. #10378–016). Luciferase assays were performed by plating $2.5 \times 10^4$ cells/well in 24 well plates. The next day, cells were co-transfected using Lipofectamine2000 with the DNA constructs indicated in each experiment in addition to *ptc*Δ136-GL3 (Chen 1999, Nybakken 2005) and pSV-Beta-galactosidase (Promega) constructs to report Hh pathway activation and normalize transfections, respectively. *Gli1* cDNA was added where relevant to activate the Hh pathway. Cells were changed to low-serum media (DMEM supplemented with 0.5% bovine calf serum and penicillin/streptomycin/glutamine) 48 hr after transfection and cultured at 37°C in 5% $CO_2$ for an additional 48 hr. Cells were harvested and luciferase and beta-galactosidase activities were measured using Luciferase Assay System (Promega, cat. # E1501) and BetaFluor β-gal assay kit (Novagen, cat. #70979–3). Multiple assays were performed and each treatment group was assayed in triplicate. Two-tailed t tests for two samples with unequal variances were used to compare samples.

## Quantitation of transgenic reporter expression data

Fluorescence data from wing confocal images were collected and quantified as previously described using the Matlab program *Icarus* (E. Ortiz-Soto, A.I.R., and S.B., manuscript in preparation) (*Parker et al., 2011*; *Ramos and Barolo, 2013*). Each experiment was performed at least two times, and fluorescence was measured from at least two wings per construct.

## EMSA competition assays

Electromobility shift competition assays were performed as previously described (*Parker et al., 2011*). EMSA oligonucleotides sequences are provided in *Figure 3—source data 1*.

## *hedgehog* misexpression in *Drosophila*

Heat shock-inducible Hh (HS-*hh*) transgenic flies (*Ingham, 1993*) were crossed with flies harboring the *ptc*LK-GFP reporter transgene. Embryos were collected overnight at 25°C and heat shocked for 1 hr at 37°C, shifted to 25°C for 30 min, and fixed, stained and imaged as described above.

## *Drosophila* larva ChIP-seq

The protocol for carrying out ChIP in larval tissue has been described previously (*Brown and Kassis, 2013*), with minor changes: fixed brains and imaginal discs were dissected from 10 third instar larvae, and before incubating the sonicated chromatin with antibodies 3.3% of each sample was saved for input reactions. ChIP was performed with 1:100 antibody dilutions of anti-Pho, anti-Ph (a kind gift from Donna J Arndt-Jovin), anti-En (Santa Cruz Biotechnologies) and 1:200 dilutions of anti-H3K27me3 (Millipore, 17–622) antibodies. Following purification of immunoprecipitated DNA, Illumina libraries were prepared by using TruSeq DNA Sample Prep Kit V2 as described previously (ethanomics.wordpress.com/chip-seq-library-construction-using-the-illumina-truseq-adapters/).     All ChIP-seq data sets were aligned using Bowtie (version 0.12.2) to the *Drosophila* reference genome (releases 5.22 and 6.02). All ChIP-seq experiments were performed with 2 biological replicates.

ChIP-seq data can be accessed at this private link for reviewers: http://www.ncbi.nlm.nih.gov/geo/query/acc.cgi?token=yrifkssspfuzbsp&acc=GSE76892.

## Mouse ChIP-seq, transgenic reporters, and beta-galactosidase staining

E10.5 embryos were obtained from timed matings of Swiss Webster (Taconic) mice and micro-dissected in cold PBS to isolate the head, trunk (neural tube and somites) and limb buds. Each tissue was pooled separately and fixed for 30 min in 1% formaldehyde/PBS at room temp. Chromatin immunoprecipitation and embryoid body differentiation were performed as previously described (*Peterson et al., 2012*). Antibodies used for ChIP were goat anti-mouse Gli2 antibody (R&D Systems, AF3635) and H3K27me3 (Millipore, 07–449). High throughput single-end sequencing was done on the GAII or Hi-seq platform (Illumina) and reads were mapped using BWA (*Li and Durbin, 2010*). Sequence data for the 500 kb flanking *Ptch1* corresponds to (chr13:63,112,841–64,166,828 (mm9)) have been deposited under accession number GSE71199. Gli3 ChIP-seq data correspond to GSE52939. Individual enhancer regions were defined by the conservation block encompassing the detected Gli binding region, amplified by PCR and cloned into a Gateway compatible version of the reporter construct previously described (*Vokes et al., 2007*). Mouse enhancer coordinates are provided in *Figure 6—source data 1*. Transient transgenic embryos were analyzed at E10.5 and X-gal stained for beta-galactosidase activity for 4 hr at 37C. ChIP-seq data are available (accession number GSE71199).

## Acknowledgements

This research was supported by the Cellular and Molecular Biology Training Grant (NIH T32-GM007315) to DSL and AIR, the University of Michigan Reproductive Sciences Training Program Fellowship to DSL, the Center for Organogenesis predoctoral fellowship (NIH T32-HD007505) to AIR and DSL, a Center for Organogenesis Research Team Grant to BLA and SB, and by NIH grant GM076509 and NSF grant MCB-1157800 to SB. APM is funded by National Institutes of Health (R01 NS033642). BSC is supported by an EBS EDGE award. BLA is funded by a Scientist Development Grant (11SDG638000) from the American Heart Association, and by a grant from the National

Institutes of Health (R01 DC014428). We thank Harold Smith (NIDDK) for Illumina NGS. SD and JAK were supported by the Intramural Research Program of the NICHD/NIH. We thank Doug Epstein, Lindy Jensen, Charles Katzman, Annie Azrak, Shelby Peterson, Autumn Holmes, Lisa Johnson, Jessica Frick, Eleanor Smith, Katherine Gurdziel and Elliott Ortiz-Soto for assistance. We thank the staff at the Microscopy and Image Analysis Laboratory (MIL) at the University of Michigan Medical School for assistance obtaining images, the members of the University of Michigan developmental genetics meeting, and the UM*fly* community for helpful discussions.

## Additional information

### Funding

| Funder | Grant reference number | Author |
|---|---|---|
| National Institute of General Medical Sciences | T32-GM007315 | David S Lorberbaum Andrea I Ramos |
| Eunice Kennedy Shriver National Institute of Child Health and Human Development | Intramural Research Program | Sandip De Judith A Kassis |
| American Heart Association | 11SDG638000 | Benjamin L Allen |
| National Science Foundation | MCB-1157800 | Scott Barolo |
| National Institute on Deafness and Other Communication Disorders | R01 DC014428 | Benjamin L Allen |
| National Institute of General Medical Sciences | R01 GM076509 | Scott Barolo |
| National Institute of Neurological Disorders and Stroke | R01 NS033642 | Andrew P McMahon |

The funders had no role in study design, data collection and interpretation, or the decision to submit the work for publication.

### Author contributions

DSL, AIR, KAP, BSC, DSP, SD, VMB, YN, MRM, SB, Conception and design, Acquisition of data, Analysis and interpretation of data, Drafting or revising the article; LEH, Acquisition of data, Analysis and interpretation of data, Drafting or revising the article; ACYC, Acquisition of data, Analysis and interpretation of data; JAK, BLA, APM, Conception and design, Analysis and interpretation of data, Drafting or revising the article

### Author ORCIDs

Matthew R McFarlane, http://orcid.org/0000-0001-8690-5633
Benjamin L Allen, http://orcid.org/0000-0003-2323-8313
Scott Barolo, http://orcid.org/0000-0002-8406-3716

### Ethics

Animal experimentation: This study was performed in strict accordance with the recommendations in the Guide for the Care and Use of Laboratory Animals of the National Institutes of Health, and with the institutional animal care protocols of Harvard University and The Jackson Laboratories, where the animal experimentation was performed. Animal husbandry and all experiments were performed in accordance with the National Institute of Health guidelines and the Institutional Animal Care and Use Committee of the University of Southern California (protocol #11867).

## Additional files

### Supplementary files

• Supplementary file 1. *patched* enhancer sequences with targeted Gli/Ci binding site mutations. Sequences were taken from the dm3 build of the UCSC genome browser. Optimal Ci binding sites are highlighted in red, low affinity Ci binding sites are highlighted in gray and further annotated with a numerical ranking corresponding to the affinity predictions previously defined (*Hallikas et al. 2006*). Blue text indicates how each Ci binding site was mutated, for example in ptc-prox enhancer DB, the optimal Ci-1 binding site of GACCACCCA was mutated to GACCaCaaCA.

• Supplementary file 2. Sequence conservation of selected TF binding motifs in enhancers of *patched*. Sequence alignments were obtained from the UCSC Genome Browser, dm6 build. Selected conserved Tango/Spineless (Tgo/Ss) motifs are in gray; Distaless (Dll) in light blue; Engrailed (En) in green; Pannier (Pnr/dGATAe) in purple; Scalloped (Sd) in blue; Cubitus interruptus (Ci) in red; Odd-Skipped (Odd) in dark red; Forkhead (Fkh) in dark yellow; Sloppy-paired (Slp) in blue-green; Pleiohomeotic (Pho) in pink, GAGA Factor (GAF/Trl) in yellow. Blocks of sequence separated by line breaks are not necessarily contiguous.

### Major datasets

The following datasets were generated:

| Author(s) | Year | Dataset title | Dataset URL | Database, license, and accessibility information |
|---|---|---|---|---|
| Peterson KA, McMahon AP | 2016 | cis-Regulatory analysis of Ptch1 | http://www.ncbi.nlm.nih.gov/geo/query/acc.cgi?token=sjwpskeknngjhun&acc=GSE71199 | Publicly available at Gene Expression Omnibus (accession no: GSE71199) |
| De S, Kassis J, Smith H | 2016 | Pho and Ph, H3K27me3 and Engrailed ChIP-seq data in Drosophila third instar larval brains and disks | http://www.ncbi.nlm.nih.gov/geo/query/acc.cgi?token=yrifkssspfuzbsp&acc=GSE76892 | Publicly available at Gene Expression Omnibus (accession no: GSE76892) |

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
