## [Decision Letter]

Thank you for submitting your article "An ancient yet flexible *cis*-regulatory architecture allows localized Hedgehog tuning by *patched/Ptch1*" for consideration by *eLife*. Your article has been reviewed by three peer reviewers, and the evaluation has been overseen by Janet Rossant as the Senior Editor and Reviewing Editor.

The reviewers are all very enthusiastic about the paper and its insights into gene regulatory architecture and want to see it published without delay. One reviewer suggested some additional transgenic experiments which could add to the paper. However, the consensus was that these were not necessary for publication, unless you happen to have the data in hand. Thus we ask that you respond to the minor comments below. We have included the comment on additional experiments for your information and comment only.

Summary:

Lorberbaum et al. present a detailed analysis of *Patched* and *Ptch1* gene regulation in flies and mice, respectively. Patched is a core component of the important hedgehog signaling pathway. It inhibits the action of Smoothened, which is required for the synthesis of activator forms of the transcriptional effectors Ci and Gli. This repression is relieved by Hedgehog-Patched interactions, leading to the induction of a variety of target genes, including Patched itself.

Past studies led to the view that Patched is regulated by simple upstream activator elements consisting of a linked Gli/Ci binding sites. Here, the authors elegantly demonstrate that this is not the actual mechanism of Patched regulation. Instead, the locus contains a series of upstream and downstream enhancers that mediate restricted induction in specific tissues at different times and in response to a variety of different signals, including Hedgehog.

This is an excellent paper. The scope of the analysis, which includes both flies and mice, is unusually broad. The findings are striking and convincing, and the analysis is thoughtful and penetrating. The paper was a pleasure to read. This type of work lays the foundation for future studies into the basis for synergy between enhancers, the relationship between binding site affinity and tissue specificity and expression level, and regulatory evolution at the locus level.

Suggested new experiments (not necessary for publication):

1) A key aspect of this article is the identification and characterization of the ptc proximal promoter (region BA) that drives tonic low levels of expression in a broad manner. This is particularly relevant because such Hh-independent expression is required to prevent ectopic pathway activation in the absence of the ligand. As this is also the case for *Ptch1*, it would be important to assay if equivalent or similar constructs containing the *Ptch1* proximal and core promoter are also able to drive low/broad expression of the reporter in mouse transgenic embryos (by lacZ staining or qPCR). The analysis of the *Ptch1* locus shown in Figure 6 is not addressing this important aspect.

2) The authors should compare what is the impact of the endogenous promoter on the expression driven by the DB element (that is, compare the pattern and levels driven by DB-hspmin and a DA reporter construct). This data could be added to Figure 3 and would address the effect of the endogenous promoter on an enhancer with clustered, high-affinity Gli motifs.

Minor points (need to be addressed):

1) In relation with Figure 3, GFP expression driven by the ptcBA transgene in the wing disc is difficult to visualize. The GFP quantification plot – currently shown in Figure 3—figure supplement 3- should be moved to the main figure.

2) It is not always clear how the different mutated constructs were made. A table with all the mutations introduced (original and mutated GBS) should be provided. For example, in Figure 5, it is not clear which motif(s) have been altered in enhancers DB, HF, VT, YU and ZY. This is important to provide for potential follow-up studies.

In addition, the sequences of the main regions tested should be provided as supplementary data and the GBS color-annotated according to predicted affinity.

3) If possible, colors should be consistent, in particular for enhancers that are described in several figures. For example, ZY is green in Figure 1, brown in Figure 5 and orange in Figure 3.

4) What is exactly the Ptc-prox construct shown in Figure 1? Does it contain a hspmin promoter?

5) Given that the nomenclature is complicated, the text should be carefully proofed for mistakes. For example, in the second to last sentence of the subsection “Bypassing enhancer-promoter specificity reveals diverse sub-patterns among segmentation stripe enhancers of *patched*” it should read LK instead of VT. The legend for Figure 3 should also be clarified, as the authors state in the text that the ptc promoter lacks a TATA box.

6) Introduction, first paragraph. It took me a few reads through these sentences to understand the contrast of patterned vs. constitutive targets. Maybe there is a way to clarify this further? Perhaps simply state that patterned targets respond in a subset of cells with an active Hh pathway, whereas universal targets respond in all Hh active cells.

A priori, there are two extreme possibilities (which is explained more clearly later in the paper). The universal pattern is either encoded by a bunch of separate enhancers, one or more for each tissue, or the universal pattern is encoded by a single regulatory element that is expressed in multiple contexts. The results of the paper demonstrate the first scenario, but it might be nice to set up these two possibilities at the outset to give some context.

7) Introduction, second paragraph. This set up was a little hard for me to follow on the first read, as it contains a lot of specific results. I would suggest a new paragraph here (starting at "We wondered[…]“, laying out what you will test beyond what is already known. First, the central question is whether the ptc(prox) enhancer sufficient to drive the universal expression pattern. Then, you hypothesize no, based on previous results where placing high affinity Gli sites into embryonic ectoderm enhancers isn't sufficient to turn them on. You then test this hypothesis by examining expression of ptc(prox) in the embryonic ectoderm.

8) Results, first paragraph. I suggest "In this study, we define a nomenclature[…] “rather than "we determined the nomenclature", which could be confused for uncovering an existing nomenclature.

9) Figure 2'-N'. What is the purpose of the greyscale half of the images? Perhaps to show the quantification more clearly? This should be clarified in the figure legend.

10) Figure 4. Using a greyscale heatmap to indicate the affinity of the predicted Gli sites is clever. But it is difficult to see on the schematics of each enhancer. You could consider using a colored background for the enhancer and superimposing the white to black sites on top of it. As long as the background is a mid-tone (not too dark and not too light), this will make the sites pop.

11) Figure 6. The expression patterns in the mouse embryos are quite difficult to see in this figure. They are small relative to the amount of space devoted to the functional genomics data. Though I can understand why it's appealing to keep them aligned, I suggest breaking the alignment to give them more space. Also, the colored thermometer shapes are not explained in the legend. Do they represent binding sites? If so, the visual convention for binding sites has changed from the heatmaps for the *Drosophila* enhancers, which is confusing. The heatmap representation conveys more information, so I suggest changing this figure to that convention in both panels.

---

## [Author Response]

Suggested new experiments (not necessary for publication):

1) A key aspect of this article is the identification and characterization of the ptc proximal promoter (region BA) that drives tonic low levels of expression in a broad manner. This is particularly relevant because such Hh-independent expression is required to prevent ectopic pathway activation in the absence of the ligand. As this is also the case for Ptch1, it would be important to assay if equivalent or similar constructs containing the Ptch1 proximal and core promoter are also able to drive low/broad expression of the reporter in mouse transgenic embryos (by lacZ staining or qPCR). The analysis of the Ptch1 locus shown in Figure 6 is not addressing this important aspect.

2) The authors should compare what is the impact of the endogenous promoter on the expression driven by the DB element (that is, compare the pattern and levels driven by DB-hspmin and a DA reporter construct). This data could be added to Figure 3 and would address the effect of the endogenous promoter on an enhancer with clustered, high-affinity Gli motifs.

One reviewer suggested some additional transgenic experiments concerning the special properties of the *ptc* promoter, but “the consensus was that these were not necessary for publication, unless you happen to have the data in hand. We have included the comment on additional experiments for your information and comment only.” Although we strongly agree that these experiments are worth performing and fully intend to do so, it’s our opinion that this analysis would be better suited to a separate follow-up report in which we can more fully characterize this promoter’s function, its interactions with a panel of enhancer elements, its chromatin states in vivo, and its regulation by PcG/TrxG.

Minor points (need to be addressed):

1) In relation with Figure 3, GFP expression driven by the ptcBA transgene in the wing disc is difficult to visualize. The GFP quantification plot – currently shown in Figure 3—figure supplement 3- should be moved to the main figure.

This has been done. See revised Figure 3, as well as revised Figure 3—figure supplement 3.

2) It is not always clear how the different mutated constructs were made. A table with all the mutations introduced (original and mutated GBS) should be provided. For example, in Figure 5, it is not clear which motif(s) have been altered in enhancers DB, HF, VT, YU and ZY. This is important to provide for potential follow-up studies.

In addition, the sequences of the main regions tested should be provided as supplementary data and the GBS color-annotated according to predicted affinity.

This has been done. We created a new Supplementary file ([Supplementary-material SD5-data]) that containsthe sequences of enhancers DB, HF, VT, YU, ZY, 1EH, GB and JB, defining each Gli binding site that was altered in our Gli binding site KO’s and affinity enhancements. We have added text references to this new file in the legends to Figure 4,Figure 5.

*3) If possible, colors should be consistent, in particular for enhancers that are described in several figures. For example, ZY is green in Figure 1, brown in Figure 5 and orange in Figure 3.*

This has been done. See revisions to all relevant figures and figure supplements (Figure 2–Figure 4, Figure 2—figure supplement 1, Figure 4—figure supplement 1, Figure 4—figure supplement 2, Figure 4—figure supplement 3, Figure 5—figure supplement 1, Figure 3—figure supplement 3) to match the colors in Figure 5.

4) What is exactly the Ptc-prox construct shown in Figure 1? Does it contain a hspmin promoter?

We added a sentence in the legend for Figure 1 indicating that we used the Hsp70“hspmin” promoter to visualize those constructs, and referenced the new [Supplementary-material SD5-data] which contains the sequences of the DB-wt and DB-CiKO constructs.

5) Given that the nomenclature is complicated, the text should be carefully proofed for mistakes. For example, in the second to last sentence of the subsection “Bypassing enhancer-promoter specificity reveals diverse sub-patterns among segmentation stripe enhancers of patched” it should read LK instead of VT. The legend for Figure 3 should also be clarified, as the authors state in the text that the ptc promoter lacks a TATA box.

This typo has been corrected, and the manuscript has been carefully proofread. Forexample, we identified a mis-labeling of an oligonucleotide in the original Figure 2—figure supplement 2, which has been corrected (those sequences are now in part B of [Supplementary-material SD5-data]).

6) Introduction, first paragraph. It took me a few reads through these sentences to understand the contrast of patterned vs. constitutive targets. Maybe there is a way to clarify this further? Perhaps simply state that patterned targets respond in a subset of cells with an active Hh pathway, whereas universal targets respond in all Hh active cells.

A priori, there are two extreme possibilities (which is explained more clearly later in the paper). The universal pattern is either encoded by a bunch of separate enhancers, one or more for each tissue, or the universal pattern is encoded by a single regulatory element that is expressed in multiple contexts. The results of the paper demonstrate the first scenario, but it might be nice to set up these two possibilities at the outset to give some context.

We have revised the text in this section to try to clarify this point (Introduction, first paragraph).

*7) Introduction, second paragraph. This set up was a little hard for me to follow on the first read, as it contains a lot of specific results. I would suggest a new paragraph here (starting at "We wondered[…]“, laying out what you will test beyond what is already known. First, the central question is whether the ptc(prox) enhancer sufficient to drive the universal expression pattern. Then, you hypothesize no, based on previous results where placing high affinity Gli sites into embryonic ectoderm enhancers isn't sufficient to turn them on. You then test this hypothesis by examining expression of ptc(prox) in the embryonic ectoderm.*

We have revised the text in this section (Introduction, third paragraph), includingreorganizing the argument along the lines suggested by the reviewer.

*8) Results, first paragraph. I suggest "In this study, we define a nomenclature[…] “rather than "we determined the nomenclature", which could be confused for uncovering an existing nomenclature.*

We have revised the text in this section (Results, first paragraph). We chose to use“we created a nomenclature[…]”, which we think accomplishes the reviewer’s goal.

*9) Figure 2. What is the purpose of the greyscale half of the images? Perhaps to show the quantification more clearly? This should be clarified in the figure legend.*

We added the line “GFP signals are isolated (greyscale) in the lower half of K'-N' to showthe lack of stripe enhancer activity in the posterior (En-positive) compartment.” to the legend of Figure 2.

10) Figure 4. Using a greyscale heatmap to indicate the affinity of the predicted Gli sites is clever. But it is difficult to see on the schematics of each enhancer. You could consider using a colored background for the enhancer and superimposing the white to black sites on top of it. As long as the background is a mid-tone (not too dark and not too light), this will make the sites pop.

We have spent a lot of time working on the best way to visuallyindicate binding motif affinity. In earlier drafts of our figures, enhancer diagrams had colored backgrounds, exactly as the reviewer suggests. However, because the indicated motifs range from dark to light, most background tones mask the medium-gray sites, which are the ones most relevant to this report (see examples, below). Also, this effect depends on the background color used, with the result that the same site looks different in different enhancers. The advantage of the white background is that, by the visual logic of the greyscale heatmap, white represents the lowest, or background, affinity. Thus Gli motifs fade into the white background as their predicted affinity decreases.

Author response image 1.**DOI:**
http://dx.doi.org/10.7554/eLife.13550.023

11) Figure 6. The expression patterns in the mouse embryos are quite difficult to see in this figure. They are small relative to the amount of space devoted to the functional genomics data. Though I can understand why it's appealing to keep them aligned, I suggest breaking the alignment to give them more space. Also, the colored thermometer shapes are not explained in the legend. Do they represent binding sites? If so, the visual convention for binding sites has changed from the heatmaps for the Drosophila enhancers, which is confusing. The heatmap representation conveys more information, so I suggest changing this figure to that convention in both panels.

We have significantly revised Figure 6 in response to these suggestions. We haveenlarged the mouse embryo images in panel A, breaking the horizontal alignment as suggested by the reviewer. We chose to use red for all Gli motifs in this figure, rather than the red+greyscale of other figures, for the sake of consistency and simplicity within this complex summary diagram. In addition to all the other information presented in these panels, indicating affinity with a range of colors would (in our opinion, after much experimentation) be visually overwhelming. However, we have followed the reviewer’s excellent suggestion and replaced the “thermometer” shapes with simple bands, consistent with the visual logic of previous figures in which binding motifs are vertical bands. These changes are reflected in revisions to the text and figure legend.